# Anxiety Modulation by Cannabinoids—The Role of Stress Responses and Coping

**DOI:** 10.3390/ijms242115777

**Published:** 2023-10-30

**Authors:** József Haller

**Affiliations:** 1Drug Research Institute, 1137 Budapest, Hungary; haller.jozsef@uni-nke.hu; 2Department of Criminal Psychology, University of Public Service, 1082 Budapest, Hungary

**Keywords:** endocannabinoid system, anxiety, laboratory studies, mechanisms, transgenic animals, pharmacology, function-enhancers

## Abstract

Endocannabinoids were implicated in a variety of pathological conditions including anxiety and are considered promising new targets for anxiolytic drug development. The optimism concerning the potentials of this system for anxiolysis is probably justified. However, the complexity of the mechanisms affected by endocannabinoids, and discrepant findings obtained with various experimental approaches makes the interpretation of research results difficult. Here, we review the anxiety-related effects of the three main interventions used to study the endocannabinoid system: pharmacological agents active at endocannabinoid-binding sites present on both the cell membrane and in the cytoplasm, genetic manipulations targeting cannabinoid receptors, and function-enhancers represented by inhibitors of endocannabinoid degradation and transport. Binding-site ligands provide inconsistent findings probably because they activate a multitude of mechanisms concomitantly. More robust findings were obtained with genetic manipulations and particularly with function enhancers, which heighten ongoing endocannabinoid activation rather than affecting all mechanisms indiscriminately. The enhancement of ongoing activity appears to ameliorate stress-induced anxiety without consistent effects on anxiety in general. Limited evidence suggests that this effect is achieved by promoting active coping styles in critical situations. These findings suggest that the functional enhancement of endocannabinoid signaling is a promising drug development target for stress-related anxiety disorders.

## 1. Introduction

### 1.1. The Endocannabinoid System

The endocannabinoid system consists of three major parts, the signaling molecules (endocannabinoids), their receptors, and the enzymatic machinery that synthesizes and degrades endocannabinoids before and after they have played their role, respectively.

Endocannabinoids are a family of lipid messengers including, but not limited to anandamide and 2-arachidonoylglycerol (2-AG). The latter were discovered about 30 years ago and were for a long time believed to be the only endogenous ligands of the cannabinoid receptors [1,2]. Subsequent research demonstrated, however, that there are several lipid messengers in the brain, which are generally called N-acylethanolamines (NAEs). These are active at various cannabinoid-binding sites, but the mechanisms activated by them only partially overlap [3].

Endocannabinoids are recognized by two G-protein coupled receptors, CB1 and CB2, which were also discovered about 30 years ago [4,5]. Originally, it was believed that the CB1 receptor is localized in the brain and affects neuronal function, whereas the CB2 is localized in the periphery and controls immunity [6]. Subsequent research, however, demonstrated that the mechanisms that mediate endocannabinoid effects are far more complex (see below).

Endocannabinoids are synthesized on demand, and after completing their role are transported back into the cytoplasm where they are degraded [7]. The routes of synthesis are complex, but there are rate-limiting enzymes by which synthesis can be influenced (e.g., NAPE-PLD and DAGLα/β for anandamide and 2-AG, respectively) [8,9]. After accomplishing their role, endocannabinoids are taken up by a carrier-mediated transport. Within the cytoplasm, anandamide is degraded via fatty-acid amide hydrolase (FAAH), whereas 2-AG is degraded via monoacylglycerol lipase (MAGL) [10].

### 1.2. Functions

Endocannabinoids are believed to modulate a wide variety of bodily and psychological functions [11]. For instance, they regulate energy metabolism [12], body temperature [13], immunity [14], fertility [15], and a wide range of other physiological phenomena. They also control almost all psychological functions, either influencing basic processes such as neurogenesis [16], neuroprotection [17] and neural energetics [18], or by putatively direct effects on anxiety, depression, cognition, reward, etc. [19,20]. In addition, the endocannabinoid system is involved in a series of pathophysiological conditions such as cancer, cardiovascular and neurodegenerative diseases [21,22,23]. This review focuses on the role played by cannabinoids in anxiety, and the therapeutic potential of agents that control the function of the endocannabinoid system.

### 1.3. Mechanisms

The mechanisms by which the endocannabinoid system achieves such wide-ranging roles are multiple, and not completely understood. It is widely believed, however, that its role in emotions and emotional behavior is primarily due to the retrograde inhibition of neuronal signaling, e.g., by postsynaptic effects on presynaptic membranes. The role of the presynaptic CB1 cannabinoid receptors in endocannabinoid actions was discovered rather early [24] and was developed later into the retrograde inhibition concept. According to this, neurotransmission elicits the postsynaptic release of endocannabinoids, which retrogradely inhibits the release of the neurotransmitter that elicited the process [25,26]. Initially it was believed that retrograde signaling is restricted to a specific type of GABAergic, particularly cholecystokinin (CCK) containing inhibitory neurons [27]. This may have alone explained the role of endocannabinoids in emotion as this type of GABA interneuron—in contrast to, e.g., parvalbumin-containing ones—alter hippocampal networks in a manner consistent with anxiolysis [27]. This assumption was later confirmed by behavioral studies demonstrating that anxiety is indeed influenced by interactions between the two systems [28]. Subsequent research showed, however, that the CB1 receptor can control a wide range of systems including glutamate, serotonin, acetylcholine, dopamine, opioid, norepinephrine, and cholecystokinin neurotransmission [28,29,30,31] (Figure 1). In addition, it was revealed that the brain expresses the CB2 cannabinoid receptor, which originally was believed to be located exclusively in the periphery [32]. Endocannabinoids also activate the postsynaptic vanilloid receptor type 1 (TRPV1) [33], or in more general terms, transient receptor potential channels (TRPV1-4; TRPA1, TRPM8; [34]. They can also affect neuronal function by the G-protein coupled receptors GPR55 and GPR18 [35,36,37]. The roles of these are poorly understood, yet they have higher Δ9-THC affinity than the classical cannabinoid receptors and are often considered actual cannabinoid receptors themselves. Endocannabinoids may also directly activate a series of intracellular signal transduction pathways [38]. Such mechanisms may be responsible for the tonic endocannabinoid signaling that controls basal synaptic neurotransmitter release, and for the interaction between neurons and glia cells, especially astrocytes [39,40].

In a way, the diversity of the molecular and cellular effects of endocannabinoids is reassuring as it explains in general terms the diversity of their role in bodily and neuronal functions. At the same time, however, it makes difficult to understand how specific functions are controlled. This review addresses the issue from the point of view of anxiety.

Firstly, we review the main research tools used to study the behavioral roles of endocannabinoid signaling. We evaluate the research questions these tools can answer regarding the control of anxiety. In the following sections, we summarize the findings obtained with each research tool. We highlight discrepancies in the literature, to evaluate the putative clinical relevance of each approach. While each contributed substantially to understanding the role of endocannabinoid signaling, their limitations differentiate them as regards their practical utility as anxiolytics. Finally, we present the main hypotheses on the anxiety-related roles of endocannabinoids. In fact, these outline the types of anxiety disorders where certain endocannabinoid agents may be useful.

## 2. Research Tools—What Questions Can They Answer?

The major tools available for the study of the endocannabinoid system in behavior can be grouped into several classes as shown below. Although each approach has its merits and advantages, they also have limitations that should be considered when research findings are interpreted.

### 2.1. Targeting Cannabinoid Receptors

Genetic tools include the disruption of the endocannabinoid receptors (CB1-KOs: [41]; CB2-KOs: [42]; CB1/CB2 double KOs: [43]) or else the induction of their overexpression in various brain regions [44,45] or in all neurons [46]. Several transgenic lines were studied, the background strains of which were different (Table 1). This may induce a certain amount of variation regarding the behavioral consequences of gene invalidation. Nevertheless, the number of genetic lines is still limited, and the very same line was used by several laboratories across the world. As such, findings can be cross-checked.

The advantage of this method is that the intervention is rather mechanism-specific. As shown above, endocannabinoids affect a variety of mechanisms. Transgenic animals offer the opportunity of studying those that are specifically mediated by either of the two receptors. In addition, knock-out mice are frequently used to elucidate the cannabinoid receptor dependence of pharmacologic agents. As such, transgenic mice are valuable tools of mechanistic studies. The disadvantage of the method is that it affects endocannabinoid signaling in the whole brain or in large brain areas in the case of conditional transgenics (see below). In addition, the invalidation of a gene brings about adaptive changes in the brain, which to a certain extent complicate the interpretation of findings.

The pharmacological equivalents of these genetic manipulations are CB1 receptor antagonists (e.g., rimonabant, the first such agent) [71] and CB1 agonists (e.g., the natural agonist delta-9 tetrahydrocannabinol [72] and CP 55,940, the first synthetic cannabinoid [73]). In recent years, a series of specific CB2 ligands were also developed [74]. See Table 2 for agents studied with respect to their anxiety-related effects. These include agonists that mimic the effects of endocannabinoids, antagonists that prevent the action of agonists including endocannabinoids, and inverse agonists which induce effects opposite to those of agonists. Inverse agonists usually also work as antagonists, especially at low doses. Agonists may be the pharmacological equivalents of gene overexpression, whereas antagonists may be the pharmacological equivalents of gene disruption. The advantages of such pharmacological methods are dual. Firstly, the effects of agents can be dosed, which cannot be achieved with genetic interventions. Secondly, pharmacological agents are more relevant clinically than genetic interventions. However, the approach also has a series of disadvantages as shown below.

It is often assumed that the tools listed above answer the question “What happens if endocannabinoid signaling is up- or downregulated?” However, this assumption is incorrect. As shown above, some endocannabinoid effects are exerted post-synaptically via the TRPV receptor [33], and TRP channels in general [34], whereas other effects are mediated by various intracellular signaling pathways [35,36,37,38]. Such cannabinoid receptor-independent effects include the modulation of neural plasticity, mitochondrial function, cardiovascular responses, etc., which do not involve cannabinoid receptors [43,84,85]. As such, the scope of research conducted with these tools should be reduced to the question “What happens if the effects of the endocannabinoid system on its receptors are inhibited or facilitated, whereas other mechanisms are unaffected or are affected in a different way”?

While the answer to this question appears useful for understanding the system, and for the development of novel pharmacological treatments, it is worth emphasizing that the direct and unconditional manipulation of cannabinoid receptors is rather unphysiological. Under physiological conditions, endocannabinoids are synthesized on demand by a Ca^2+^-dependent mechanism [86]. Consequently, endocannabinoid signaling inhibits neurotransmission where this takes place and if it crosses a certain threshold [87,88]. By contrast, exogenously administered cannabinoid ligands inhibit neurotransmission indiscriminately because the receptors are present continuously in large amounts throughout the brain [89], moreover, throughout the whole organism [90]. These receptors are affected concomitantly irrespective of the momentary release of endocannabinoids. Instead of limiting or augmenting ongoing neurotransmission by a homeostasis-like process, ligands affect neurotransmission independently of the natural requirements of neural function. In addition, such unphysiological alterations in cannabinoid signaling can perturb neural connections and, consequently, the cortical oscillations associated with physiological functions [91]. In plain terms, endocannabinoid functions are targeted, whereas the exogenous up- and downregulation of their receptors is pervasive.

These considerations naturally do not invalidate findings obtained by methods that genetically or pharmacologically target endocannabinoid receptors, but one should reformulate the question that such studies can answer. “What happens if cannabinoid receptors are nonspecifically up- or downregulated at the level of the whole brain?” or in the case of local administrations “at the level of rather large brain regions”. This in turn generates a second question: “To what extent such non-physiological manipulations reflect the physiological functions of endocannabinoid signaling”?

### 2.2. Targeting the Degrading Enzymes

The considerations presented above are not valid to endocannabinoid signaling alone but to pharmacology overall. Neural communication is targeted (except for extrasynaptic or volumetric neurotransmission), whereas pharmacologic agents have general effects in the brain. However, the endocannabinoid system stands out as it modulates a wide range of neurotransmitter and non-neurotransmitter systems, which aggravates the problems of poor coupling to physiological functioning. Fortunately, there is an alternative to direct receptor modulation, which circumvents the problem to a certain extent. In particular, one can target the enzymes that degrade endocannabinoids to terminate the signal. One such enzyme, FAAH, is primarily active on anandamide, but can also degrade other endocannabinoids such as 2-AG and oleamide [92]. The enzyme is widely present in the organism. In the brain, the CB1 receptor and the FAAH enzyme are located in a complementary fashion; FAAH is preferentially located post-synaptically to CB1, by which it can control the access of the receptor to anandamide [93]. MAGL is more specific than FAAH, although in addition to 2-AG it also catalyzes some non-endocannabinoid fatty acids [94]. In contrast to FAAH, MAGL colocalizes with the CB1 receptor pre-synaptically, which is also an appropriate place for controlling the 2-AG supply of the receptor [95]. The two enzymes may partially colocalize, at least in the hippocampus [96].

The first enzyme inhibitor developed was URB-597, which inhibits the enzyme FAAH [97]. FAAH inhibition increases the brain levels of anandamide by prolonging the duration of the endocannabinoid signal. Another compound, JZL 184, inhibits the enzyme MAGL, which degrades 2-AG and enhances retrograde signaling by this major endocannabinoid [98]. In addition to these two inhibitors, others were also studied in anxiety tests (see Table 3). The FAAH enzyme was also studied by invalidating its gene [99] and by inducing its overexpression by viral gene transfer [100]. MAGL-KO mice were also created [101], but the over-expression of its gene has not yet been achieved to our knowledge.

The main advantage of such agents is that they are more activity-bound than receptor ligands. Endocannabinoids are synthesized on demand and are rapidly degraded after their release [7,87]. This means that cannabinoids are present in the synapse for short periods only. The enzymes have no substrates between bouts of activation; therefore, their inhibition is inconsequential when the endocannabinoid system is inactive. This implies that they alter physiological responses rather than induce non-physiological responses. The two inhibitors also allow the separation of the effects of the two main endocannabinoids, which is impossible with cannabinoid ligands as these bind to the same receptors. However, this seems to be of little importance in the case of anxiety, as the consequences of FAAH and MAGL inhibition appear similar (Table 3), even though differences were observed in other behaviors [121]. The question that may be asked by the application of such agents is “What happens if the natural endocannabinoid signaling was prolonged or if it was shortened”? In plain terms, enzyme manipulations maintain the targeted nature of the physiological process.

### 2.3. Targeting Other Mechanisms

The selective inhibition of enzymes responsible for the biosynthesis of endocannabinoids show the advantages of those agents that target degrading enzymes [122]. To our knowledge, however, the anxiety-related effects of such agents were only sporadically studied so far and consequently will not be discussed here.

Another physiologically sound way of studying the endocannabinoid system is the blockade of their membrane transport (e.g., by AM-404) [123]. The termination of the signal necessitates the transfer of endocannabinoids from the extracellular space into the cytoplasm as the degrading enzymes are located intracellularly. This is achieved by several membrane proteins, the most important being the endocannabinoid membrane transporter [124,125]. This is widely present in neuronal membranes but also in peripheral organs [126]. In principle, this pharmacological tool answers the same question as the one that can be answered by using enzyme inhibitors. The anxiety-related effects of AM-404 were less well-studied than those of enzyme inhibitors, but the available data will be reviewed in Section 3.3.

## 3. Cannabinoids and Anxiety

### 3.1. Transgenic Animals

According to the general view, the invalidation of the Cnr1 gene that encodes the CB1 receptor increases anxiety. This suggests that the CB1 receptor is involved in neuronal processes that decrease anxiety. Although this assumption is supported by a relatively large number of studies, it does not seem to be entirely true (Table 1).

Out of 26 studies found in PubMed after a careful search, CB1-deficient mice proved to be anxious in 17 reports (65% of all studies), and whilst only 1 study found that CB1 gene disruption decreased anxiety, there were 8 (31%) in which the same manipulation had no effect. As such, only a minority of studies reported “non canonical” findings, but this minority was by far non-negligible. Similarly discrepant findings were obtained with the CB2 receptor. What is the reason for such discrepancies?

It occurs that neither the background strain of gene manipulation nor the anxiety test employed explain this variation in findings, except perhaps for the shock-prod burying test. Albeit the test was validated pharmacologically for anxiety, it is quite different from the rest of the tests and elicits complex behavioral responses which may explain the anxiolytic-like effect of CB1 gene disruption in its case. Apart from this, however, discrepant findings were obtained with all background strains and all anxiety tests. Although the life-long absence of the KO gene could have caused developmental adaptations, this may have had little impact on the discrepancies between findings because such adaptations were likely similar especially when the background strain was similar, moreover, the same genetic line was used by conflicting reports.

In addition to the role of gender and age (one study each), it emerges that the aversiveness of the testing environment played a role in supporting the anxiogenic effects of CB1 gene disruption, whereas decreased aversiveness appeared to abolish this effect. Aversiveness in this context is defined as a fearful condition under which the anxiety test was performed. Such aversive conditions include high light (light intensity levels around or above 800 lx), which is aversive to nocturnal animals like rodents; the unfamiliarity of animals with the experimenters or the testing environment; unexpected changes in the environment during testing, etc. For example, ref. [52] shows that the consequences of CB1 gene disruption depended largely on the aversiveness of the test arena illumination in this study. We provide more details on aversiveness in Section 3.3, because this condition was studied more systematically with such agents than with transgenic animals. Another report suggested that CB1 gene invalidation affected anxiety-like behavior by altering coping strategies, which also depends on test aversiveness [127]. The significance of similar findings will be outlined in Section 3.4.

Taken together, the data seem to suggest that the invalidation of the Cnr1 gene increases anxiety conditionally, and the condition to this effect is the aversiveness of the testing environment. One can hypothesize that the role of aversiveness is greater than that transpiring from the data of Table 1. Anxiety tests are performed in laboratory-specific conditions, which may involve hidden aversive components. These are sometimes difficult to identify from the Methods sections of publications but may be revealed by information provided by authors on request. Such hidden aversive conditions will be detailed in the section on FAAH inhibition.

In addition to studies on whole-brain (whole-organism) gene disruption, there are studies on conditional knockouts where the Cnr1 gene was invalidated in specific cell types and/or specific brain areas. Such studies, believed to reveal the mechanisms of gene disruption in more detail, are incontestably highly valuable, but they are not devoid of discrepancies either.

Firstly, local gene manipulations identified many different mechanisms, the comparison or ranking of which is difficult. For instance, a series of studies suggested that forebrain glutamatergic neurons have a major role in the anxiogenic effects of general CB1 gene disruption, as CB1 disruption in this area increased anxiety to a similar magnitude than that seen in CB1-KOs [53,56]. In addition, the rescue of CB1 receptors on dorsal telencephalic glutamatergic neurons abolished anxiogenic effects observed upon CB1 receptor loss [58]. However, similarly high anxiety was elicited by the disruption of CB1 receptors in the medial septum and the nucleus of the diagonal band [128], and in the dopaminergic neurons of the ventral tegmental area [129]. As such, we have three widely different local mechanisms that may all underlie the effects of general Cnr1 gene disruption. Based on these findings, one can hypothesize that anxiety increases irrespective of the brain location or the neurochemical mechanism of gene disruption. However, this is unlikely.

Secondly, the number of discrepant findings is large with local manipulations. For instance, CB1 receptor disruption in cortical glutamatergic neurons did not affect anxiety in one study [52]. In addition, the overexpression of the CB1 receptor in the median prefrontal cortex increased anxiety in the social interaction test [45], which is in contrast with the anxiogenic effect of CB1 disruption in this area. Somewhat contrasting findings were also obtained with the CB2 receptor. Their disruption in midbrain dopaminergic neurons decreased anxiety [130,131], whereas their general disruption was anxiogenic. With the CB1 receptor, the effects of general and dopamine-specific gene disruptions were similar, whereas with the CB2 they were dissimilar.

Taken together, studies in transgenic mice suggest that the elimination of CB1 or CB2 receptors in the whole brain or parts of it may or may not increase anxiety. In the case of CB1-KOs, the factor that differentiates effect from no effect appears to be the aversiveness of the testing environment. However, the role of this was only studied specifically in a restricted number of studies. In the case of conditional transgenic animals, discrepancies likely arise from the differential role of the affected brain regions in anxiety. Cannabinoid receptors are expressed in a variety of brain regions and cell types, many of which play opposite roles in anxiety. As such, discrepant findings may be natural consequences of this variation in roles. The issue may be solved by a more comprehensive approach, e.g., by systematic comparative studies into the main anxiety-related brain areas and neurotransmitter systems.

Complete CB1 disruption was not reported in humans so far, yet polymorphisms in the gene encoding the receptor were shown to affect anxiety. For instance, polymorphisms in the promoter region of the gene increased vulnerability to anxiety disorders in conjunction with polymorphisms in the promoter regions of the serotonin transporter [132]. It was also shown that genetic variability in the promoter and coding region of the CB1 gene affected the extinguishing of learned fear [133], and such polymorphisms conferred vulnerability to panic disorder in females [134]. Taken together, these findings suggest that polymorphisms in the CB1 gene underlie individual differences in human anxiety, which indirectly supports the laboratory findings with knockout animals.

### 3.2. CB1 Receptor Pharmacology

Many cannabinoid receptor ligands were synthesized since the development of CP 55,940 and rimonabant. A recent list of CB1 ligands contains 56 entries [135], and the list is not complete as new synthetic cannabinoids are regularly synthesized for recreational purposes [136]. Many from this large number of ligands were studied for their effects on anxiety; therefore, an exhaustive review of the literature would be far beyond the scope of this study. Instead, we will focus on findings that outline the main features of this research area.

In theory, the effects of CB1 gene disruption and CB1 antagonists should be similar, yet this is not always so. For instance, the invalidation of the CB1 gene increased whereas the CB1 inverse agonist rimonabant (SR141716) decreased anxiety in the wild types of KO mice [48]. Moreover, rimonabant decreased anxiety in CB1-KOs as well, suggesting an effect independent of the CB1 receptor. Although a subsequent study showed convergent effects with the CB1 antagonist AM-251 [49], a series of similar findings led to the conclusion that cannabinoids have an unknown third receptor. As one of the studies of the time suggested “Recent evidence suggests that a third CB3 receptor is out there, waiting to be cloned” [130]. The CB3 receptor concept was later discarded, but subsequent research found multiple non-CB1/non-CB2 action sites for cannabinoids (see Section 1.3 “Mechanisms”). Nevertheless, these early studies were right in assuming that the CB1 and the CB2 receptors are not sufficient to explain the effects of cannabinoids. Indeed, cannabinoid ligands bind to a variety, moreover, different set of targets, which results in differences in their behavioral profile, including but not limited to anxiety [137,138]. In addition, CB1 agonists and antagonists may affect the same phenomena by different mechanisms. For instance, the antagonist AM-251 and rimonabant are direct antagonists of mu-opioid receptors, whereas the CB1 agonist WIN-55212-2 affects pain perception via interactions between the CB1 and opioid receptors [139].

The divergent pharmacological profile of CB1 ligands leads to a series of discrepant findings, a few of which are outlined in Table 2.

In blunt terms, both CB1 agonists and antagonists may either promote or inhibit anxiety or may leave it unaffected. One and the same antagonist (e.g., rimonabant) increased and decreased anxiety in two studies each, whereas three other CB1 antagonists failed to affect anxiety. All three agonists, inverse agonists, and antagonists (Δ9-THC, rimonabant and AM-251, respectively) increased anxiety, whereas three different agonists had biphasic effects. Such findings are no strong arguments for the statement that CB1 ligands are promising targets of anxiolytic drug development.

Various hypotheses were advanced to explain such discrepant findings. One of the obvious explanations is that different ligands have different impacts on the mechanisms that mediate cannabinoid effects. For instance, the ability of cannabinoids to affect the GPR55 receptor is variable; moreover, it may depend on the cell type and the tissue [140]. In addition, one and the same agent, particularly AM-251, is an antagonist of the CB1 but an agonist of the GPR55, which is also considered a cannabinoid receptor. Similarly, the ability of cannabinoid ligands to exert effects through transient receptor potential (TRP) channels is also highly variable; moreover, ligands exert variable effects on various subtypes of such receptors (TRPV1-4; TRPA1, TRPM8; [34]). The CB1 agonist WIN55212-2 increased intracellular calcium in cell cultures, while the agonist CP 55,940 did not [141]. These findings suggest that cannabinoid ligands have individual spectrums as it regards the mechanisms they can activate. This may account for both the differential effects of putatively similar agents (e.g., CB1 antagonists) and the biphasic effects of cannabinoid agonists. Regarding the latter, one can hypothesize that in parallel with the increase in dosage, more and more mechanisms are activated, which changes the behavioral outcome of the treatment.

Even the effects of cannabinoids on the CB1 and CB2 receptors may show large variations due to a differential brain distribution. It was shown that the brain distribution of the cannabinoid ligands WIN-55,212 and SR-141716A is not uniform [142]. For instance, two times more WIN-55,212 was accumulated in the hypothalamus than in the amygdala after injecting the compound intraperitoneally. The accumulation of SR-141716A was also inhomogeneous and importantly, the accumulation preferences of the two compounds were different. The highest levels of SR-141716A were found in the prefrontal cortex, whereas the lowest was observed in the cerebellum, which is different from that seen with WIN-55,212 (highest: hypothalamus; lowest: amygdala). Although the issue is insufficiently studied, this report suggests that cannabinoids have a “brain fingerprint” of distribution, with obvious consequences for behavior. One can hypothesize that in parallel with increasing the dosage, cannabinoids can reach the threshold for effect in more and more brain areas, which can change the behavioral outcome of the treatment.

These considerations may make the development of receptor ligand anxiolytics rather difficult. A clinically useful anxiolytic should selectively target brain areas where endocannabinoids affect anxiety favorably; moreover, they should target the appropriate neuron types within this brain area, and out of the multitude of intracellular binding sites should specifically affect those that have favorable effects on anxiety. This would involve the optimization of the new compound for all three, brain distribution, cell-, and molecular mechanism-selectivity. Drug development at this level of complexity is close to hopeless.

It is worth noting that the effects of pharmacologic agents also depend on stress exposure like those of CB1 gene disruption. For instance, the biphasic effect of HU-210 on anxiety under normal conditions lost its biphasic nature in chronically stressed animals [81]. Although there is a tight interaction between the mechanisms of the stress response and endocannabinoid signaling [143], the impact of stress on anxiolytic efficacy was less well-studied with receptor ligands than with CB1-KOs or enzyme inhibitors (see below), except for pain-induced anxiety.

A specific case of anxiety associated with stress is that elicited by chronic pain, which activates nociceptive afferents that target brain regions involved in affective and cognitive processes [144]. Consequently, chronic pain is frequently associated with anxiety. Pain perception is mediated by neurotransmitter systems that are subject to retrograde control by endocannabinoid signaling, e.g., GABA, norepinephrine, TRPV1, etc. [144,145,146]. Not surprisingly, the favorable impact of cannabinoid receptor ligands on pain-induced anxiety is more robust than in the case of other anxiety types [147]. Even Δ9-THC provided positive findings, although the applicability of this compound is hampered by its side-effects. Nevertheless, recent advances improved the human applicability of this compound as shown below, and in addition, more natural ways of enhancing endocannabinoid signaling are also effective (see Section 3.3 “Enzyme inhibitors and transporters”).

The complex effects of cannabinoid receptor ligands may also explain the controversial effects of natural cannabinoids in humans. Chronic consumption of cannabis either increased or decreased anxiety, depending on the study [148,149]. Such conflicting findings may be due to individual differences in cannabinoid receptor distribution or genetic polymorphisms in humans, but it transpires that one major factor relates to the composition of cannabis preparations. In addition to Δ9-THC, cannabis contains a variety of cannabinoids and other psychoactive compounds, including cannabidiol (CBD) [150]. The latter does not share the psychotropic effects of Δ9-THC and has the opposite neural and behavioral effects [151]. This is valid also for the anxiety-related effects of cannabis, as products high in Δ9-THC but low in CBD proved to be anxiogenic whereas those having a low THC:CBD ratio were anxiolytic [152]. This assumption is also supported by a review showing that no human studies provided evidence of anxiolytic effects of isolated Δ9-THC, whereas isolated CBD reliably decreased anxiety in a variety of studies [153]. The finding that CBD can counteract many unwanted effects of Δ9-THC including anxiety led to the assumption that products that contain large amounts of CBD, or if CBD was added to cannabis, may make cannabis consumption safer [154]. Indeed, a variety of studies showed that CBD can counteract the anxiogenic effects of Δ9-THC when mixtures of the two compounds were consumed [155,156,157]. Similar findings were obtained with various preparations, especially with naboxilols (e.g., Sativex) in which the Δ9-THC:CBD ratio is approximately 1:1. The absolute amount Δ9-THC and the baseline anxiety of participants is also relevant for the favorable effects of CBD [155,158]. The creation of safer cannabinoid preparations is naturally important, especially for consumers of younger ages, where Δ9-THC may induce long-term increases in anxiety and may contribute to the development of anxiety disorders [159,160,161]. In addition, the stable Δ9-THC-CBD ratio of naboxilols makes them applicable for therapeutic purposes. For instance, naboxilols were shown to decrease anxiety during cannabis withdrawal [162,163]. In addition, THC:CBD mixtures supported withdrawal, probably because of its CBD content. Such mixtures may also be applicable in conditions associated with pain. There are various conditions where chronic pain results in chronic stress and anxiety, and these may be ameliorated by treatments involving endocannabinoid signaling [164,165]. One option would be the application of naboxilols or cannabis preparations high in cannabidiol content, which lack the anxiogenic effects of Δ9-THC [166]. Other cannabinoid signaling-related treatments are also applicable (see Section 3.3).

It is important to note, however, that the addition of CBD to cannabinoid preparations does not make Δ9-THC an anxiolytic compound, whereas CBD, although present in cannabis, is not a primary ligand of CB1 and CB2 receptors [167]. As such, discussing its anxiety-related effects in detail is outside the scope of this review. In conclusion, cannabis may activate mechanisms involved in a reduction in anxiety, yet the multitude of mechanisms activated by this compound—like with other cannabinoid ligands—make it an unreliable anxiolytic agent, whereas Δ9-THC per se appears to be anxiogenic in humans.

### 3.3. Enzyme Inhibitors and Transporters

When we started to study the effects of the FAAH inhibitor URB-597 on anxiety, there was a wealth of literature on the issue, most of which reported positive findings. The compound reduced anxiety in a variety of species and strains, and in a variety of anxiety tests (Table 3A). One study showed that similar effects can be obtained by invalidating the gene of the FAAH enzyme [104], whereas another showed that the overexpression of FAAH in the prefrontal cortex increased anxiety [100]. These studies showed that FAAH inhibition is anxiolytic, suggesting that the enzyme is a promising target for drug development.

However, URB-597 did not affect anxiety in several studies (Table 3A); in addition, FAAH-KO mice were not different from controls in another [106].

Such discrepancies are usual in pharmacological research in general and also in endocannabinoid research. However, it was reported earlier that URB-597 considerably ameliorated the corticosterone response and amygdala activation elicited by restraint stress [168,169]. This suggested that FAAH inhibition worked as a buffer against environmental aversiveness. This can be considered a likely consequence of its main molecular effect, e.g., the enhancement of anandamide signaling. Stress-induced and pathological anxiety is believed to result from excessive neuronal (especially glutamatergic) activation [170,171]; therefore, a reduction in this activation by retrograde endocannabinoid inhibition would logically reduce anxiety as well. In support of the involvement of stress, a modified—more stressful—version of the elevated plus maze test revealed anxiolytic effects in a study where FAAH inhibition did not affect anxiety under normal conditions [106]. Prompted by this publication, we inspected earlier reports, and interviewed authors. The information gathered suggested that testing was aversive in all studies where URB-597 was effective [108]. In some cases, this was part of the experimental protocol, e.g., when anxiety was studied during alcohol withdrawal [103]. In other cases, aversiveness derived from local experimental practices. For instance, the testing arena was brightly illuminated, or the light source was placed such that the closed arms were shadowed whereas the open arms were illuminated in the elevated plus-maze test. In other cases, the study was performed in animals housed individually, or in animals that were not handled and not habituated to the testing environment. We showed earlier that aversive testing conditions enhanced the effects of CB1 gene disruption on anxiety [51] and hypothesized that the same was true for FAAH inhibition.

The hypothesis on the role of aversiveness was supported by the findings. URB-597 did not affect anxiety when the stressfulness of the testing environment was minimized, but decreased anxiety under three conditions: when animals were not handled and were not habituated to the testing environment, when they were tested under high light, and when the illumination of the experimental room underwent sudden changes [108,119]. In contrast to FAAH inhibition, the anxiolytic effect of the benzodiazepine chlordiazepoxide was not changed by these conditions. Taken together, this suggested that FAAH inhibition has no specific effect on anxiety but abolishes the anxiogenic effects of aversive environments. Similar findings were obtained in a variety of stress paradigms and FAAH inhibitors (Table 3B). Interestingly, highly similar results were obtained with the MAGL inhibitor JZL-184 that enhances retrograde signaling by 2-AG (Table 3C). The first study on the issue [118] entirely replicated earlier findings with URB-597 [108], but the endocannabinoid that counteracted the anxiogenic effects of aversive testing environments was 2-AG in this case.

There have been a few studies which studied the two enzyme inhibitors comparatively. Two studies found that the effects of MAGL inhibition were more robust than the effects of FAAH inhibition [113,115], whereas the third found the reverse [114]. As such, both enzymes may become targets of anxiolytic drug development. However, chronic MAGL inhibition was shown to desensitize central CB1 receptors and to produce other unwanted effects. For instance, the life-long invalidation of the MAGL gene chronically elevated brain 2-AG levels, desensitized CB1 receptors in brain regions involved in the control of emotional states, and enhanced excitatory drive in the basolateral amygdala-medial prefrontal (mPFC) circuit, with subsequent elevation of glutamate release to the mPFC [172]. Not surprisingly, these transgenic animals showed anxiety and obsessive–compulsive behaviors in the light/dark box and marble burying tests [172]. CB1 receptor desensitization was also observed in mice chronically treated with the MAGL inhibitor JZL184, but not in those chronically treated with the FAAH inhibitor PF-3845 [173]. The unwanted effect profile of the two inhibitors showed similar differences in other studies as well [174,175]. These findings suggest that in the long term, MAGL inhibition involves risks that are not shared by chronic FAAH inhibition, even though in the short run, both interventions decrease stress-induced anxiety. These findings render FAAH a better target for anxiolytic drug development than MAGL.

The cannabinoid transport blocker AM-404 replicated the effects of enzyme inhibitors. This compound decreased anxiety in a variety of paradigms (e.g., the step-down avoidance task, Vogel conflict test, contextual fear, elevated plus-maze, defensive withdrawal, and separation-induced ultrasonic vocalizations) [176,177,178,179,180]. Only one study found that AM-404 did not decrease anxiety [77], but this result could have been confounded by environmental influences. Indeed, similar to FAAH inhibition, AM-404 ameliorated endocrine and neuronal responses to stress [168,169] and one study found that its effect depended on the stress history of subjects [181]. It is noteworthy that many of the tests where the compound was effective implicitly involved aversiveness.

The favorable stress-related effects of the agents discussed here potentially render them useful in psychopathological states elicited by traumatic experience, e.g., post-traumatic stress disorder [182]. Indeed, FAAH inhibitors not only ameliorated stress-induced anxiety (Table 3B,C) but also promoted the extinction of aversive memories in several paradigms [183,184,185]. Like enzyme inhibitors, the endocannabinoid transport inhibitor AM404 also facilitated the extinction of conditioned fear [186].

Interestingly, similar findings were obtained in humans, when fear extinction was studied in conjunction with gene variants of the FAAH enzyme. In one study, a single-nucleotide polymorphism associated with lower catabolic performance of FAAH increased plasma anandamide levels and changed brain activation patterns elicited by a fear-conditioning paradigm [113]. Although the genotype did not affect fear extinction, an indirect favorable effect was observed, this being mediated by the increase in anandamide levels and their impact on brain activation patterns. In another study, genetic variation within the FAAH gene influenced physiological, cognitive, and neural signatures of fear learning in women with PTSD [114], whereas a subsequent study published by an overlapping set of authors showed that the inherited FAAH deficit decreased anxiety responses elicited by extinction recall [187]. Finally, FAAH inhibitors administered to humans elicited effects similar to those of existing anxiolytic agents and dampened brain responses to emotional stimuli [188]. Moreover, FAAH inhibition improved the recall of fear extinction memories and attenuated the anxiogenic effects of stress [189]. FAAH inhibitors may also be useful in conditions involving chronic pain (e.g., cancer, fibromyalgia) stress, and anxiety [190,191]. Taken together, these findings point to the translational power of laboratory studies and render FAAH a promising drug target for stress-induced anxiety disorders.

### 3.4. FAAH Inhibitors and Coping Styles

A relatively new line of evidence suggests that FAAH inhibition goes beyond the amelioration of stress-induced anxiety by promoting a change in coping styles. The idea of addressing this issue came from studies suggesting that URB-597 promoted active coping in both the forced swimming [192], and the fear conditioning tests [127]. In fact, these authors recorded the usual behavioral variables, but interpreted them in terms of coping styles rather than in terms of depression- and anxiety-like behavior. Fortunately, coping styles can be investigated in tests that are associated neither with anxiety nor with depression, e.g., the tail pinch and back tests (see below). Employing such tests may allow the separation of coping responses from effects on anxiety and depression.

Active and passive coping styles are two distinct behavioral phenotypes which differ in the way challenges are dealt with [193,194]. In this context, challenges are in fact stressors and aversive conditions that need to be handled when they occur. Active copers attempt to control challenges when they occur (problem-oriented coping) whereas passive copers respond to challenges by avoidant behavior. These temporally stable behavioral phenotypes have adaptive significance in both animals and humans [193,194,195,196].

The rationale behind this line of research was that anxiety-like behavior is rather akin to passivity in laboratory tests, whereas anxiolysis is usually identified by an active response. For instance, the avoidance of the potentially dangerous open arms in the elevated plus maze indicates anxiety, and at the same time it is also indicative of passive coping, as the challenge imposed by the open arms is avoided. In a similar fashion, investigation of the open arms is an active coping strategy, which at the same indicates decreased anxiety. The same is true for most tests that investigate depression-like behavior. Floating in the forced swimming test indicates a passive response to the challenge of forced swimming and is a sign of depression-like behavior. Likewise, trials to escape the situation (struggling) are an active response and at the same time a sign of reduced depression. An effect of coping styles may be considered a common denominator of effects on anxiety and depression and may explain both effects of endocannabinoid signaling.

In the first study addressing the issue, we employed the tail-pinch test. This simple tests consists in attaching a clamp to the tail of rats [197]. Passive copers endure the clamp-induced discomfort by disregarding it, whereas active copers try to discard the clamp [198]. The distribution of these two coping strategies was bimodal, e.g., mixed strategies were less frequent than either active or passive coping. FAAH inhibition by URB-597 promoted active coping, as the share of rats adopting the active strategy almost doubled, mostly at the expense of passive copers. In a follow-up study, we studied mice, which were forced on their backs [199]. Responses to this forced unnatural position are indicative of coping styles [200]. Behavior showed bimodal distribution in the back test: mice either showed escape attempts or equally distributed time between passivity and escape. URB-597 increased escapes in animals with low escape scores. In the same study, URB-597 promoted active responses in the fear-conditioning paradigm, where in addition to decreasing freezing, it increased locomotion [199].

This approach to the interpretation of FAAH inhibition effects received little attention so far, although similar findings were obtained in humans. It was shown for instance that stress-coping traits are predicted by FAAH gene-mediated differences in amygdala threat processing [201]. In this study, carriers of a low-expressing FAAH variant exhibited quicker habituation of amygdala reactivity to threat and had lower scores on the personality trait of stress-reactivity. As such, low FAAH activity was associated with low stress responsiveness as a trait, and a quicker habituation to threatening stimuli.

## 4. Overall Interpretation of the Findings

The findings reviewed above justify the interest in the anxiolytic potential of compounds that target the endocannabinoid system. It occurs that the enhancement in endocannabinoid signaling decreases anxiety in a variety of paradigms. The least robust findings were obtained with cannabinoid receptor ligands, which influence the whole organism from immunity to cardiovascular functions [14,23]. This is possible because cannabinoid ligands control the function of eight neurotransmitter systems pre-synaptically [28,29,30,31] and of TRP channels post-synaptically [33,34]; in addition, they influence multiple intracellular signaling pathways [35,36,37,38] and modulate interactions between glia cells and neurons (Figure 2) [39,40]. Many or all these mechanisms are affected by both cannabinoid agonists and antagonists. In addition, one and the same agent may be an antagonist at one of the targets while being an agonist at another; moreover, the spectrum of effects is compound-specific and depends on the type of cells that are exposed to the ligands [140]. Cannabinoid ligands do affect mechanisms involved in anxiety control, yet the multitude of functions affected in parallel appears to translate as conflicting findings in research.

From a mechanistic point of view, genetic manipulations led to more reliable findings, likely because their effects are restricted—in theory at least—to just one of the multitude of mechanisms that mediate the effects of endocannabinoids. Although secondary changes—in fact, adaptations—may and do occur, the invalidation or the overexpression of the CB1 receptor, for instance, is incomparably more specific than the inactivation or activation of the whole system by receptor ligands. In addition, specificity can be increased by targeted interventions that focus on one brain area, or one neurotransmitter system. Not surprisingly, discrepant findings are fewer with transgenic animals than with pharmacologic agents, and even these discrepancies may be explained by the dependence of consequences on stress exposure.

Enzyme and transporter inhibitors are the most specific interventions available at present. Such agents amplify ongoing endocannabinoid activations instead of influencing the whole system. The chronic administration of particular agents may lead to unwanted secondary changes, but acutely, these agents provide the most valuable information on the functions of the endocannabinoid system and are probably the most promising targets for anxiolytic drug development.

Three hypotheses have been put forward so far on the role of endocannabinoids in emotional and behavioral control.

The endocannabinoid system controls behavior by its interactions with the stress system (HPA-axis) and ensures normal functioning by eliminating excessive stress responses at all three levels: the hormonal, neural, and behavioral [202].The endocannabinoid system contributes to the integration of perception and execution, by allowing this adaptation to the environment. It buffers maladaptive responses and protects against psychiatric symptoms [203].The endocannabinoid system promotes an active coping with challenges, which confers the organism advantages in critical situations. This may be the common denominator of its anxiolytic- and antidepression-like effects [204].

These three hypotheses appear complementary rather than contradictory. All three suggest that the endocannabinoid system is a valid target for the treatment of psychiatric conditions associated with dysregulated affect. The task is to find the agent that achieves the goal with minimal risks.

## Figures and Tables

**Figure 1 ijms-24-15777-f001:**
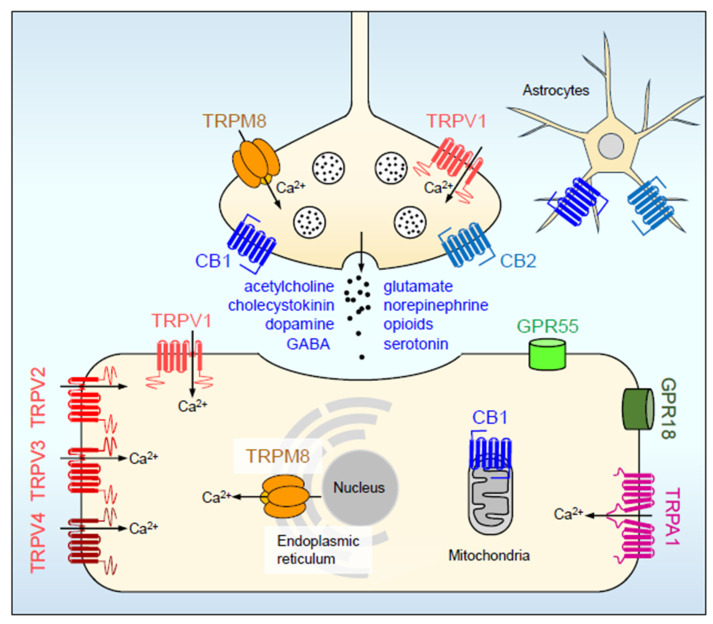
Target diversity of cannabinoid receptor ligands. These bind to, and affect the function of, presynaptic, postsynaptic, membrane and intracellular receptors and ion channels. Receptor ligands retrogradely inhibit 8 neurotransmitter systems, influence extracellular and intracellular cation exchanges (primarily Ca^2+^), have an impact on the energy metabolism of neurons and mediate communication with glia cells. The spectrum of effects is cannabinoid ligand-specific and depends largely on the cell types and their physiological state. For details see text.

**Figure 2 ijms-24-15777-f002:**
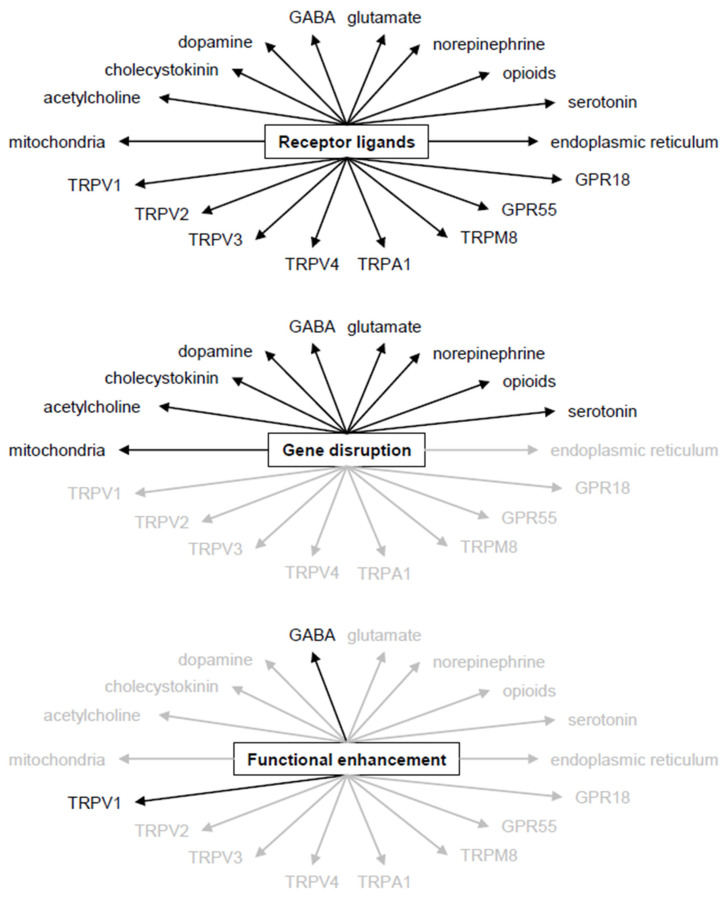
Target selectivity of the main research approaches. (**Upper panel**) In theory, cannabinoid receptor ligands activate all mechanisms that are controlled by cannabinoids. Practically, however, ligands may bind to a subgroup of the target molecules only, due to the ligand specificity of mechanisms. Nevertheless, each ligand activates a multitude of mechanisms, even if not all of them. (**Middle panel**) The disruption of cannabinoid receptor genes eliminates responses mediated by the receptors but allows effects mediated by other mechanisms. (**Lower panel**) Functional enhancers (inhibitors of degrading enzymes and of transport proteins) affect only those cannabinoid functions that are activated by the situation. This reduces the number of mechanisms affected. For details see text. Black, mechanism activated; grey, mechanism not affected.

**Table 1 ijms-24-15777-t001:** The effects of CB1 gene disruption on anxiety.

**CB1-KO**			
**Background (sex)**	**Effects on anxiety**	**Anxiety test**	**Reference**
CD1 (♂)	↑	LD	[47]
CD1 (♂)	↑	EPM	[48]
CD1 (♂)	↑	EPM	[49]
CD1 (♂)	↑ (increased stress response)	EPM, LD, SI	[50]
CD1 (♂)	↑ (high light)	EPM	[51]
C57BL/6NCrl (♂)	↑	EPM, LD	[52]
CD1 (♂)	↑ (0.7, 1.5 mA)	CF	[53]
ICR (♂)	↑ (chronic stress-like phenotype)	EPM	[54]
CD1 (♂)	↑ (increased stress response)	EPM	[55]
CD1 (♂)	↑ (chronic stress)	CF	[56]
C57/BL6J (♂)	↑	SI	[57]
C57BL/6J (♂)	↑	EPM, LD	[58]
C57BL/6J (♂)	↑	EPM, LD	[59]
C57BL/6N (♂, ♀)	↑	SI	[60]
CD1 (♂)	↑ (young)	OF, LD	[61]
C57BL/6J (♂, ♀)	↑ (chronic pain)	EZM, LD	[62]
C57BL/6J (♂)	↑ (male)	EPM	[63]
CD1 (♂)	→ (old)	OF, LD	[61]
CD1 (♂)	→ (low light)	EPM	[51]
CD1 (♂)	→	EPM, OF	[64]
CD1 (♂)	→ (unstressed)	CF	[56]
CD1 (♂)	→ (0.5 mA)	CF	[53]
CD1 (♂)	→	SI	[65]
C57BL/6J (♂, ♀)	→ (control)	EZM, LD	[62]
C57BL/6J (♀)	→ (female)	EPM	[63]
C57BL⁄6 (♂)	↓	SPBT	[66]
**CB2-KO**			
**Background**	**Effects on anxiety**	**Anxiety test**	**Reference**
C57BL/6J (♂)	↑	EPM, LD	[67]
C57BL/J6 (♀)	↑	OF	[68]
C57BL/6J (♂, ♀)	→	EZM, OF	[69]
**Gene Overexpression**			
**Receptor**	**Effects on anxiety**	**Anxiety test**	**Reference**
CB1 * (♂)	↑	SI	[45]
CB2 (♂)	↓	EPM, LD	[70]

*Changes in anxiety*. ↑, increased; →, unaltered; ↓, decreased. *Sex*. ♂, male; ♀, female. *Anxiety tests*. CF, conditioned fear; EPM, elevated plus-maze; EZM, elevated zero maze; LD, light/dark box; OF, open field; SI, social interaction; SPBT, shock prod burying test. *, restricted to the medial prefrontal cortex. Note that the condition under which the gene manipulation was effective or ineffective was indicated in brackets in the case of studies where transgenic mice were studied under more than one set of conditions.

**Table 2 ijms-24-15777-t002:** Discrepant findings with CB1 receptor ligands—examples.

**The Effects of CB1 Antagonists Is Conflicting**			
**Antagonist**	**Effect on Anxiety**	**Effective Dose (Range)**	**Reference**
Rimonabant	anxiogenesis (EPM, ETM, OF)	3 (1–3) mg/kg	[64]
	anxiolysis (EPM)	10 (1–10) mg/kg	[75]
	anxiogenesis (EPM, DW)	3 (0.1–3) mg/kg	[76]
	anxiolysis (EPM)	3 (1–3) mg/kg	[48]
AM-281	no effect (LD)	none (1–4) mg/kg	[77]
AM4113	no effect (EPM)	none (3.0–12.0) mg/kg	[78]
AVE1625	no effect (LD)	none (10–100) mg/kg	[79]
**Agonists and Antagonists May Have Similar Effects**			
**Ligand**	**Effect on Anxiety**	**Effective Dose (Range)**	**Reference**
Δ9-THC	anxiogenesis (EPM)	1, 2.5, 10 (0.25–10) mg/kg	[80]
Rimonabant	anxiogenesis (EPM)	3, 10 (1–10) mg/kg	
AM-251	anxiogenesis (EPM)	3, 10 (1–10) mg/kg	
**Agonists Have Biphasic Effects**			
**Ligand**	**Effect on Anxiety**	**Effective Dose (Range)**	**Reference**
HU-210	anxiolysis (EPM)	10 μg/kg	[81]
	anxiogenesis (EPM)	50 μg/kg	
CP 55,940	anxiolysis (EPM)	1 μg/kg	[82]
	anxiogenesis (EPM)	50 μg/kg	
Δ9-THC	anxiolysis (LD, OF)	0.2 mg/kg	[83]
	anxiogenesis (LD, OF)	7.5 mg/kg	

All studies were performed in males. EPM, elevated plus-maze; ETM, elevated T-maze; LD, light/dark box; OF, open field; DW, defensive withdrawal.

**Table 3 ijms-24-15777-t003:** The impact of degrading enzymes on anxiety.

**A. Early Studies with URB-597**			
**Effective dose (range)**	**Species (test; sex)**	**Effects on anxiety**	**Reference**
0.1 (0.05–0.1) mg/kg	rat (EZM; ♂)	↓	[97]
0.1, 0.3 (0.03–0.3) mg/kg	mouse (EPM; ♂)	↓	[80]
0.1, 0.3 (0.1–0.3) mg/kg	mouse (EPM; ♀)	↓	[102]
0.3, 1 (0.3–1) mg/kg	rat (EPM; ♂)	↓	[103]
1 (1) mg/kg	mouse (EPM; ♂)	↓	[104]
0.1, 0.3 (0.1–0.3) mg/kg	rat (LD; ♂)	↓	[105]
none (0.1–10) mg/kg	mouse (EPM; ♂, ♀)	→	[106]
none (0.03–0.3) mg/kg	mouse (EPM; ♂)	→	[107]
1 (1) mg/kg	mouse (LD; ♂)	→	[104]
**B. FAAH Inhibition Ameliorates Stress-Induced Anxiety**			
**FAAH inhibitor**	**Species**	**Stress factor**	**Reference**
URB-597	mouse (EPM; ♂, ♀)	light contrast *	[106]
URB-597	rat (EPM; ♂)	test aversiveness	[108]
URB-597	rat (EPM, SPBT; ♂)	nicotine withdrawal	[109]
JNJ-5003	mouse (EPM; ♂)	chronic stress	[110]
PF-3845	mouse (LD, NH; ♂)	contextual fear	[111]
OL-135	rat (FRE; ♂)	contextual fear	[112]
PF-3845	rat (LD; ♂)	chronic stress	[113]
URB-597	rat (FRE, SI; ♂)	contextual fear	[114]
PF-3845	rat (EPM; ♂)	alcohol withdrawal	[115]
URB-597	rat (IA; ♂)	contextual fear	[116]
URB-597	rat (FRE, ASR; ♂)	contextual fear	[117]
**C. MAGL Inhibition Ameliorates Stress-Induced Anxiety**			
**MAGL inhibitor**	**Species**	**Stress factor**	**Reference**
JZL-184	rat (EPM; ♂)	test aversiveness	[118]
JZL-184	mouse (EPM; ♂)	hormone manipulation	[119]
JZL-184	mouse (LD, NH; ♂, ♀)	Susceptibility ^†^	[120]
JZL-184	rat (LD; ♂)	chronic stress	[113]
JZL-184	rat (FRE, SI; ♂)	contextual fear	[114]
JZL-184	rat (EPM; ♂)	alcohol withdrawal	[115]

↓, anxiety decreased; →, anxiety not changed; ASR, acoustic startle response; EPM, elevated plus-maze; FRE, freezing in shock-associated environment; IA, inhibitory avoidance; LD, light–dark box; NH, novelty-induced hypophagia; SI, social interaction test; SPBT, shock probe burying test; ♂,male; ♀, female; *, the open arms of the EPM brightly illuminated, closed arms remained in shadow; ^†^, groups were selected based on stress susceptibility.

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
