# Peer review of "Anxiety Modulation by Cannabinoids—The Role of Stress Responses and Coping"

_ijms, 2023, doi:10.3390/ijms242115777_

Round 1

Reviewer 1 Report

Comments and Suggestions for Authors

The manuscript is interesting and well-structured. The author provided a critical review of published articles investigating how manipulations of the endocannabinoid system affect anxiety-like behaviors in animal models. 3 main types of manipulations are described: transgenic mouse models with altered cannabinoid receptor expression; pharmacologic targeting of endocannabinoids receptors, and pharmacologic targeting of endocannabinoids degrading enzymes or other mechanisms. Discrepancies between findings are discussed in light of the methodological limitations that could have led to confounding results, and possible interpretations of these discrepancies are reported. An interesting paragraph discusses how the anxiolytic effect of FAAH inhibitors could be linked to their ability to induce a switch from passive to active coping strategies.

I have a few suggestions:

line 40-41: please provide a reference for “as this type of GABA interneurons – in contrast to, e.g., parvalbumin-containing ones – alter hippocampal networks in a manner consistent with anxiolysis”

line 72-74: I suggest rephrasing, e.g. “It is often assumed that the tools listed above answer the question “What happens if endocannabinoid signaling is up- or downregulated?" However, this assumption is incorrect.”

Table 1: the data reported and discussed are useful and informative. However, the message could have been made more apparent in the table by a simple reorganization. I suggest listing mice of the same background one after the other, as well as placing close together studies that used the same test. For example, it would be very useful to have the two lines referring to {45} one after the other, to make the effect of stress (high vs low light) immediately apparent.

line 169-174: Please refrain from comments on other authors’ choice of title. The contradiction is not as evident as claimed in the manuscript. The work described in [49] shows that depletion of CB1 affects anxiety in a way that depends on the aversiveness of the environment.  The title is not misleading since saying “ (proper functioning of ) Endocannabinoids render exploratory behavior largely independent of the test aversiveness” is not that different form saying “Altered functioning of the endocannabinoid system render exploratory behavior largely dependent of the test aversiveness”. Therefore I suggest rephrasing to avoid comments that go behond discussion often data.

For example, instead of “This was true even for a study stating in the title that “Endocannabinoids render exploratory behavior largely independent of the test aversiveness” [46]. Despite the title, the consequences of CB1 gene disruption depended largely on the aversiveness of test arena illumination in this study. As such, there was a contradiction between the title and the content of the study.”

Use:

“For example, [46] shows that the consequences of CB1 gene disruption depended largely on the aversiveness of test arena illumination in this study. Another report…”

In the “transgenic animals” session it would be useful to add a few lines mentioning that, in constitutive KO mice, the life-long absence of the KO gene could have caused developmental adaptations, potentially contributing to discrepancies between findings. Although this is briefly alluded to in the “Overall interpretation of the findings”, I believe the review can benefit from bringing this up sooner and more clearly.

line 335: in “The hypothesis was supported by the findings.” please specify which findings.

line 378-379: in “Yet, coping styles can be investigated in tests that are associated neither with anxiety nor with depression.” It would be nice to have a reference to the test mentioned, even if they are described later on in the paragraph.

Lines 388-389. in “An effect on coping styles may be considered a common denominator of these seemingly different responses” it is unclear what the “seemingly different responses” are.

Other points:

line 10-12: This is an important point. I suggest splitting the sentence in 2 to make it easier to read and stand out more. for example “The optimism concerning the potentials of this system for anxiolysis is probably justified. However, the complexity of the mechanisms affected by endocannabinoids, and discrepant findings obtained with various experimental approaches makes the interpretation of research results difficult.”

line 31: substitute “incompletely understood” with “not completely understood”

Line 47: in the sentence “the postsynaptic vanilloid receptor type 1 (TRPV1) receptor” I think the second “receptor” should be removed.

Line 53: please remove the “etc.”

line 68, line 250, and others. The “e.g.” should be placed into brackets throughout the text to facilitate reading. For example, in line 68:  instead of “receptor antagonists, e.g., rimonabant the first such agent” use “receptor antagonists (e.g. rimonabant, the first such agent),”. Line 250: use  “(e.g., rimonabant). etc..

Line 167: can older age itself be considered a factor increasing the adverseness of the external environment?

Line 252-253: this list is confusing “All three the CB1 agonist Δ9-THC and the CB1 inverse agonists and antagonists rimonabant and AM-251 increased anxiety”. My suggestion is to restructure the list to make it clearer and add punctuation.

Lines 264-265: please rephrase, such as: ”The CB1 agonist WIN55212-2 increased intracellular calcium in cell cultures, while the agonist CP 55940 did not [85]”.

Line 289-91: I suggest rephrasing the sentence “The individual spectrum of brain distribution and mechanism activation of cannabinoids may require sophisticated and complex studies, which may or may not be worthwhile.” One possibility may be “Identifying the activation mechanism of cannabinoids and how it changes across brain region would require sophisticated and complex studies that may not be worth pursuing”.

Line 301: I suggest removing “with”.

Line 348: substitute “, which” with “that”

line 355-358: Please change the sentence from “This compound decreased anxiety in a variety of paradigms, in e.g., the step-down avoidance task ….vocalizations [80, 111-115]” to “This compound decreased anxiety in a variety of paradigms (e.g. the step-down avoidance task ….vocalizations) [80, 111-115]”

Line 358-361: I believe this part could have been written more clearly, for example, instead of “Albeit ineffective in one study [77], this may have been due to environmental influences. Similar to FAAH inhibition, AM-404 ameliorated endocrine and neuronal responses stress to stress [107,108] and in one study where this was explicitly studied, its effect depended on the stress history of subjects [116].”

It could have been written on the line of “Only one study found that AM-404 did not decrease anxiety [77], but this result could have been confounded by environmental influences. Indeed, similar to FAAH inhibition, AM-404 ameliorated endocrine and neuronal responses stress to stress [107,108] and one study found that its effect depended on the stress history of subjects [116].”

Line 372-364: I suggest a slight change in text from “A relatively new line of evidence suggests that the consequences of FAAH inhibition go beyond the amelioration of stress-induced anxiety. Particularly, it was shown that it can change coping styles” to “A relatively new line of evidence suggests that the FAAH inhibition goes beyond the amelioration of stress-induced anxiety by promoting a change in coping styles”

The molecule URB-597 is sometimes written as URB597 or URB.597. Please stick to one form.

Line 419-422: this sentence “Cannabinoid ligands control the function of 8 neurotransmitter systems presynaptically, TRP channels postsynaptically, the communication between glia cells and neurons, and affect a series of intracellular signaling pathways by which they affect the functioning of the whole organism from immunity to cardiovascular functions.” is difficult to read because too long.

I suggest splitting it in 2, such as “Cannabinoid ligands influence the whole organism from immunity to cardiovascular functions. This is possible because cannabinoid ligands control the function of 8 neurotransmitter systems presynaptically and of TRP channels postsynaptically, as well as modulating the communication between glia cells and neurons, and influencing multiple intracellular signaling pathways.”

Comments on the Quality of English Language

The review is in my opinion well done. I'm not a native English speaker, so I can't in good faith evaluate in detail the use of the English language. However, I think that sometimes the manuscript uses sentences that are too long and shows poor use of punctuation.  Thus, the text is sometimes not easy to read. My suggestion is to ask the opinion of a native English speaker.

Author Response

Reply to Reviewer 1

We thank the reviewer for her/his helpful and expert comments. We believe that the paper was substantially improved by observing them. Please find below my replies to the comments with a detailed account of the changes made. Note that changed text was highlighted by blue font in the manuscript.

Comment No. 1. line 40-41: please provide a reference for “as this type of GABA interneurons – in contrast to, e.g., parvalbumin-containing ones – alter hippocampal networks in a manner consistent with anxiolysis”.

Reply. In fact, the reference to the previous sentence applied to this one as well. Nevertheless, we added a new reference that confirmed assumptions based on experimental work (see lines 75-77).

Comment No. 2. line 72-74: I suggest rephrasing, e.g. “It is often assumed that the tools listed above answer the question “What happens if endocannabinoid signaling is up- or downregulated?" However, this assumption is incorrect.”

Reply. Done (lines 138-140). Thank you for reformulating the sentence.

Comment No. 3. Table 1: the data reported and discussed are useful and informative. However, the message could have been made more apparent in the table by a simple reorganization. I suggest listing mice of the same background one after the other, as well as placing close together studies that used the same test. For example, it would be very useful to have the two lines referring to {45} one after the other, to make the effect of stress (high vs low light) immediately apparent.

Reply. I could make this change if the reviewer insisted, but I think arranging studies based on the direction of the effect (anxiogenesis – no effect – anxiolysis) conveys the message of the table better.

Comment No. 4. line 169-174: Please refrain from comments on other authors’ choice of title. The contradiction is not as evident as claimed in the manuscript. The work described in [49] shows that depletion of CB1 affects anxiety in a way that depends on the aversiveness of the environment.  The title is not misleading since saying “ (proper functioning of ) Endocannabinoids render exploratory behavior largely independent of the test aversiveness” is not that different form saying “Altered functioning of the endocannabinoid system render exploratory behavior largely dependent of the test aversiveness”. Therefore I suggest rephrasing to avoid comments that go behond discussion often data. For example, instead of “This was true even for a study stating in the title that “Endocannabinoids render exploratory behavior largely independent of the test aversiveness” [46]. Despite the title, the consequences of CB1 gene disruption depended largely on the aversiveness of test arena illumination in this study. As such, there was a contradiction between the title and the content of the study.” Use: “For example, [46] shows that the consequences of CB1 gene disruption depended largely on the aversiveness of test arena illumination in this study. Another report…”

Reply. Text was changed as suggested (lines 263-265).

Comment No. 3. In the “transgenic animals” session it would be useful to add a few lines mentioning that, in constitutive KO mice, the life-long absence of the KO gene could have caused developmental adaptations, potentially contributing to discrepancies between findings. Although this is briefly alluded to in the “Overall interpretation of the findings”, I believe the review can benefit from bringing this up sooner and more clearly.

Reply. Although the life-long absence of the KO gene could have caused developmental adaptations, this may have had little impact on the discrepancies between findings because such adaptations were likely similar when the background strain was similar, moreover, when the same genetic line was used by conflicting reports. As such, we do not think that such adaptations contributed to the discrepancies between findings. Nevertheless, the issue was mentioned (lines 251-255).

Comment No. 5. line 335: in “The hypothesis was supported by the findings.” please specify which findings.

Reply. The new version specifies that the hypothesis concerns the role of aversiveness (line 439).

Comment No. 6. line 378-379: in “Yet, coping styles can be investigated in tests that are associated neither with anxiety nor with depression.” It would be nice to have a reference to the test mentioned, even if they are described later on in the paragraph.

Reply. The tests were named early on as suggested by the reviewer (lines 496-497).

Comment No. 6. Lines 388-389. in “An effect on coping styles may be considered a common denominator of these seemingly different responses” it is unclear what the “seemingly different responses” are.

Reply. The new version clarifies that we referred to anxiety and depression (line 513-515).

Comment No. 7. line 10-12: This is an important point. I suggest splitting the sentence in 2 to make it easier to read and stand out more. for example “The optimism concerning the potentials of this system for anxiolysis is probably justified. However, the complexity of the mechanisms affected by endocannabinoids, and discrepant findings obtained with various experimental approaches makes the interpretation of research results difficult.”

Reply. Thank you for this suggestion. It was observed (lines 9-12)

Comment No. 8. line 31: substitute “incompletely understood” with “not completely understood”

Reply. Corrected (line 64)

Comment No. 9. Line 47: in the sentence “the postsynaptic vanilloid receptor type 1 (TRPV1) receptor” I think the second “receptor” should be removed.

Reply. Corrected (superfluous “receptor” deleted).

Comment No. 10. Line 53: please remove the “etc.”

Reply. “etc.” removed.

Comment No. 11. line 68, line 250, and others. The “e.g.” should be placed into brackets throughout the text to facilitate reading. For example, in line 68:  instead of “receptor antagonists, e.g., rimonabant the first such agent” use “receptor antagonists (e.g. rimonabant, the first such agent),”. Line 250: use  “(e.g., rimonabant). etc..

Reply. Brackets were added throughout.

Comment No. 12. Line 167: can older age itself be considered a factor increasing the adverseness of the external environment?

Reply. Possibly, but the issue may be too complex to be considered here.

Comment No. 13. Line 252-253: this list is confusing “All three the CB1 agonist Δ9-THC and the CB1 inverse agonists and antagonists rimonabant and AM-251 increased anxiety”. My suggestion is to restructure the list to make it clearer and add punctuation.

Reply. The sentence was simplified (lines 352-353)

Comment No. 14. Lines 264-265: please rephrase, such as: ”The CB1 agonist WIN55212-2 increased intracellular calcium in cell cultures, while the agonist CP 55940 did not [85]”.

Reply. Sentence changed as suggested (lines 364-365).

Comment No. 15. Line 289-91: I suggest rephrasing the sentence “The individual spectrum of brain distribution and mechanism activation of cannabinoids may require sophisticated and complex studies, which may or may not be worthwhile.” One possibility may be “Identifying the activation mechanism of cannabinoids and how it changes across brain region would require sophisticated and complex studies that may not be worth pursuing”.

Reply. Thank you for the sentence. However, the paragraph was reorganized on the suggestion of Reviewer 2 (lines 385-391).

Comment No. 16. Line 301: I suggest removing “with”.

Reply. Done.

Comment No. 17. Line 348: substitute “, which” with “that”

Reply. Done.

Comment No. 18. line 355-358: Please change the sentence from “This compound decreased anxiety in a variety of paradigms, in e.g., the step-down avoidance task ….vocalizations [80, 111-115]” to “This compound decreased anxiety in a variety of paradigms (e.g. the step-down avoidance task ….vocalizations) [80, 111-115]”

Reply. Done (lines 476-478)

Comment No. 19. Line 358-361: I believe this part could have been written more clearly, for example, instead of “Albeit ineffective in one study [77], this may have been due to environmental influences. Similar to FAAH inhibition, AM-404 ameliorated endocrine and neuronal responses stress to stress [107,108] and in one study where this was explicitly studied, its effect depended on the stress history of subjects [116].” It could have been written on the line of “Only one study found that AM-404 did not decrease anxiety [77], but this result could have been confounded by environmental influences. Indeed, similar to FAAH inhibition, AM-404 ameliorated endocrine and neuronal responses stress to stress [107,108] and one study found that its effect depended on the stress history of subjects [116].”

Reply. Thank you. Sentence corrected as suggested (lines 479-482).

Comment No. 20. Line 372-364: I suggest a slight change in text from “A relatively new line of evidence suggests that the consequences of FAAH inhibition go beyond the amelioration of stress-induced anxiety. Particularly, it was shown that it can change coping styles” to “A relatively new line of evidence suggests that the FAAH inhibition goes beyond the amelioration of stress-induced anxiety by promoting a change in coping styles”

Reply. Thank you. Sentence improved as suggested (lines 472-478).

Comment No. 21. The molecule URB-597 is sometimes written as URB597 or URB.597. Please stick to one form.

Reply. Now I consistently used URB-597 throughout.

Comment No. 22. Line 419-422: this sentence “Cannabinoid ligands control the function of 8 neurotransmitter systems presynaptically, TRP channels postsynaptically, the communication between glia cells and neurons, and affect a series of intracellular signaling pathways by which they affect the functioning of the whole organism from immunity to cardiovascular functions.” is difficult to read because too long. I suggest splitting it in 2, such as “Cannabinoid ligands influence the whole organism from immunity to cardiovascular functions. This is possible because cannabinoid ligands control the function of 8 neurotransmitter systems presynaptically and of TRP channels postsynaptically, as well as modulating the communication between glia cells and neurons, and influencing multiple intracellular signaling pathways.”

Reply. Thank you. The sentence was changed as suggested (lines 557-561).

Comment No. 25. The review is in my opinion well done. I'm not a native English speaker, so I can't in good faith evaluate in detail the use of the English language. However, I think that sometimes the manuscript uses sentences that are too long and shows poor use of punctuation.  Thus, the text is sometimes not easy to read. My suggestion is to ask the opinion of a native English speaker.

Reply. I thank the reviewer for his/her language corrections. All were observed.

Reviewer 2 Report

Comments and Suggestions for Authors

The article “Anxiety modulation by cannabinoids-the role of stimulus responding and coping” covers a significant and relevant topic related to the endocannabinoid system and its effects on anxiety and coping styles. The findings presented in this study have potential implications for drug development in psychiatric disorders, which adds to its importance. Overall, the study presents promising results, but it requires significant improvements to meet the high standards of the journal. However, the manuscript requires major revisions to enhance its scientific rigor and readability.

Major comments:

1.     The abstract provides a concise overview of the study's objectives and main findings. However, it would be beneficial to include specific information on the research methods and primary results to give readers a better understanding of the review's scope. Additionally, mention of the significance or potential implications of the findings for anxiolytic drug development would enhance the abstract.

2.     Give examples of specific inhibitors of endocannabinoid degradation and transport that have been utilized in studies related to anxiety.

3.     The introduction effectively introduces the broad functions of endocannabinoids, but it would be helpful to provide a clear and focused statement of the review's primary aim and research question. Specify whether the review aims to highlight discrepancies in the literature, provide a comprehensive overview, or identify potential areas for further research.

4.     When discussing the mechanisms of the endocannabinoid system, it is essential to cite relevant studies or experimental evidence to support the claims made. Including references will enhance the credibility of the review.

5.     Consider providing a brief overview of the main classes of endocannabinoids and their receptors (CB1 and CB2) to orient readers who might be less familiar with the subject matter.

6.     Towards the end of the introduction, it would be beneficial to include a clear transition sentence that bridges the discussion of general endocannabinoid functions to its specific role in anxiety. This will help readers understand the relevance and significance of the review's focus.

7.     Consider expanding the description of CB1 and CB2 receptor knockout (KO) animal models and overexpression approaches, as well as their relevance to studying anxiety. Mention any specific anxiety-related behaviors or phenotypes observed in these models, if available.

8.     Elaborate on the pharmacological agents, such as CB1 receptor antagonists and agonists, used to mimic genetic manipulations. Clarify their mechanisms of action and discuss how their use can influence the interpretation of study results.

9.     When discussing the unphysiological alterations resulting from exogenous ligand administration, provide references or evidence supporting these claims to bolster the argument.

10.  Provide more background information on the role of fatty-acid amide hydrolase (FAAH) and monoacylglycerol lipase (MAGL) in endocannabinoid degradation and regulation of anxiety-related processes.

11.  If possible, include any insights into the distinct roles of anandamide and 2-arachidonoylglycerol (2-AG) in anxiety modulation, considering the use of FAAH and MAGL inhibitors.

12.  Mention specific studies that have investigated the effects of FAAH and MAGL inhibitors on anxiety-related behaviors to support the statement that they maintain targeted physiological processes.

13.  Elaborate on the mechanism of action of AM-404 in inhibiting endocannabinoid membrane transport, and discuss its potential advantages and limitations compared to enzyme inhibitors. Provide relevant references to support the claims made.

14.  If there are any available studies on the anxiety-related effects of AM-404, include a brief summary of their findings.

15.  Consider adding a concise introductory paragraph that sets the context for the section, highlighting the importance of transgenic animal studies in understanding the role of cannabinoid receptors in anxiety.

16.  Clarify whether the 65% increase in anxiety observed in CB1-deficient mice is a statistically significant trend or if individual studies varied widely in their results.

17.  Table 1: Include a brief explanation of the specific anxiety tests used (e.g., CF, EPM, EZM, LD, OF, SI, SPBT) to help readers understand the methods used in the studies.

18.  When discussing the variation in findings for CB1-deficient mice, provide possible explanations for the contradictory results (e.g., differences in experimental conditions, testing protocols, genetic background) to shed light on the reasons for the discrepancies.

19.  Provide a clear definition of "aversiveness" in the context of anxiety tests to ensure readers understand its significance in the interpretation of results.

20.  Expand on the studies that investigated the role of aversiveness in anxiety regulation, highlighting their methodologies and findings. Discuss the implications of these findings for interpreting the anxiolytic effects of CB1 gene disruption under different test conditions.

21.  What do the subsequent chapters mean?

22.  Summarize the main findings and implications of the discrepancies observed in transgenic animal studies regarding the effects of CB1 and CB2 receptor disruptions on anxiety. Offer insights into potential future research directions that could help resolve conflicting results.

23.  When discussing the possible existence of a "CB3 receptor" based on early studies, clarify that subsequent research found multiple non-CB1/non-CB2 action sites for cannabinoids, and there is no known CB3 receptor. Emphasize the importance of understanding these non-CB1/non-CB2 targets to explain the divergent pharmacological profile of CB1 ligands.

24.  Provide additional studies or evidence supporting the concept of a "brain fingerprint" for cannabinoid distribution and its impact on behavior. Discuss how this non-uniform brain distribution can influence anxiety regulation.

25.  Elaborate on the potential challenges and limitations associated with developing drugs targeting specific brain regions and cannabinoid mechanisms for anxiolytic effects. Discuss the complexities of such drug development and the feasibility of achieving desired outcomes.

26.  When discussing the initial positive findings with URB-597 and FAAH-KO mice, clarify the role of FAAH in the metabolism of anandamide, leading to increased anandamide levels and potential anxiolytic effects.

27.  Elaborate on the mechanism by which FAAH inhibition acts as a buffer against environmental aversiveness, leading to anxiolytic effects. Discuss the role of stress and aversive testing conditions in modulating the effects of FAAH inhibition on anxiety.

28.  When comparing the effects of FAAH and MAGL inhibitors, provide more context regarding the desensitization of central CB1 receptors with chronic MAGL inhibition and the absence of such effects with FAAH inhibitors. This will help readers understand the potential advantages and disadvantages of each approach for anxiolytic drug development.

29.  Elaborate on the mechanism by which AM-404 affects anxiety, specifically focusing on its role as a transporter blocker and its impact on the availability of endocannabinoids in the brain.

30.  Discuss the potential therapeutic applications of FAAH and AM-404 inhibitors in the treatment of psychopathological states induced by traumatic experiences, such as post-traumatic stress disorder (PTSD). Elaborate on the evidence supporting their effectiveness in promoting fear extinction and ameliorating stress-induced anxiety.

31.  Discuss how these coping styles may influence responses to stress and aversive situations.

32.  Explain the rationale behind investigating coping styles as a common denominator of anxiety-like behavior and anxiolysis. Discuss how traditional anxiety tests may primarily assess passive coping behaviors, while anxiolytic effects may involve active coping responses.

33.  Include a brief section discussing human studies that support the link between FAAH gene variants, coping traits, and threat processing in the amygdala. Emphasize the potential relevance of these findings for understanding individual differences in stress response and coping strategies.

34.  The statement that "cannabis consumption increases anxiety" needs to be supported by specific references to the studies mentioned in the review (e.g., studies performed over the last 40 years [134-137]). Providing more context and evidence for this claim would enhance the credibility of the interpretation.

Minor comments:

1.     Check for consistency in verb tenses throughout the manuscript.

2.     Review sentence structure and ensure proper subject-verb agreement.

3.     Check for typographical errors and punctuation mistakes.

4.     Ensure accurate and consistent citation formatting throughout the manuscript.

Comments on the Quality of English Language

Author Response

Reply to Reviewer 2

We thank the reviewer for her/his helpful and expert comments. We believe that the paper was substantially improved by observing them. Please find below my replies to the comments with a detailed account of the changes made. Note that changed text was highlighted by dark red in the manuscript.

Comment No. 1. The abstract provides a concise overview of the study's objectives and main findings. However, it would be beneficial to include specific information on the research methods and primary results to give readers a better understanding of the review's scope. Additionally, mention of the significance or potential implications of the findings for anxiolytic drug development would enhance the abstract.

Reply. We rewrote the abstract in line with this comment (see lines 16-24)

Comment No. 2. Give examples of specific inhibitors of endocannabinoid degradation and transport that have been utilized in studies related to anxiety.

Reply. Please note that the number of words is limited to 200 in the abstract. This precludes naming all the agents that were studied; naming a few would be unfair. All agents were presented in the text.

Comment No. 3. The introduction effectively introduces the broad functions of endocannabinoids, but it would be helpful to provide a clear and focused statement of the review's primary aim and research question. Specify whether the review aims to highlight discrepancies in the literature, provide a comprehensive overview, or identify potential areas for further research.

Reply. We added a paragraph to Introduction to address this comment. This paragraph outlines the structure and the aims of the review (lines 93-101).

Comment No. 4. When discussing the mechanisms of the endocannabinoid system, it is essential to cite relevant studies or experimental evidence to support the claims made. Including references will enhance the credibility of the review.

Reply. The Introduction was completed with a series of new references (see outlined references in Introduction).

Comment No. 5. Consider providing a brief overview of the main classes of endocannabinoids and their receptors (CB1 and CB2) to orient readers who might be less familiar with the subject matter.

Reply. In the new version, Introduction starts with a section, which briefly describes the endocannabinoid system (lines 30-52).

Comment No. 6. Towards the end of the introduction, it would be beneficial to include a clear transition sentence that bridges the discussion of general endocannabinoid functions to its specific role in anxiety. This will help readers understand the relevance and significance of the review's focus.

Reply. I added a new paragraph to the end of Introduction in response to Comment 3. I believe that this paragraph fulfils the role of the bridging sentence suggested here.

Comment No. 7. Consider expanding the description of CB1 and CB2 receptor knockout (KO) animal models and overexpression approaches, as well as their relevance to studying anxiety. Mention any specific anxiety-related behaviors or phenotypes observed in these models, if available.

Reply. Section “2.1. Targeting cannabinoid receptors” was expanded to address the issues mentioned by the Reviewer (lines 110-123).

Comment No. 8. Elaborate on the pharmacological agents, such as CB1 receptor antagonists and agonists, used to mimic genetic manipulations. Clarify their mechanisms of action and discuss how their use can influence the interpretation of study results.

Reply. As with Comment 7, the section “2.1. Targeting cannabinoid receptors” was expanded to address the issues suggested by the Reviewer (lines 127-137).

Comment No. 9. When discussing the unphysiological alterations resulting from exogenous ligand administration, provide references or evidence supporting these claims to bolster the argument.

Reply. The section was enriched with new references (see dark red reference marks, lines 148-163)

Comment No. 10. Provide more background information on the role of fatty-acid amide hydrolase (FAAH) and monoacylglycerol lipase (MAGL) in endocannabinoid degradation and regulation of anxiety-related processes.

Reply. Background information on FAAH and MAGL were provided (lines 179-188)

Comment No. 11. If possible, include any insights into the distinct roles of anandamide and 2-arachidonoylglycerol (2-AG) in anxiety modulation, considering the use of FAAH and MAGL inhibitors.

Reply. We specified in the new version that there are no differences in the anxiety-related effects of FAAH and MAGL inhibitors, even though the two compounds affected other behaviors differentially. As such, the separation of anandamide and 2-AG effects is less important with anxiety than with other behaviors (lines 205-207).

Comment No. 12. Mention specific studies that have investigated the effects of FAAH and MAGL inhibitors on anxiety-related behaviors to support the statement that they maintain targeted physiological processes.

Reply. We added two sentences and 2 references to support the claim (lines 198-201).

Comment No. 13. Elaborate on the mechanism of action of AM-404 in inhibiting endocannabinoid membrane transport, and discuss its potential advantages and limitations compared to enzyme inhibitors. Provide relevant references to support the claims made.

Reply. The new version provides summary background information on the endocannabinoid transporter (lines 217-222).

Comment No. 14. If there are any available studies on the anxiety-related effects of AM-404, include a brief summary of their findings.

Reply. The anxiety-related effects of AM-404 were presented already in the earlier version (lines 471-479 in revisions; note that this part of the text was not outlined because it was present in the original submission). To be clearer, we specified in section “2.3. Other mechanisms” that the anxiety related effects of AM-404 are presented in section “3. Cannabinoids and anxiety” (3.3. Enzyme inhibitors and transporters) (lines 223-225).

Comment No. 15. Consider adding a concise introductory paragraph that sets the context for the section, highlighting the importance of transgenic animal studies in understanding the role of cannabinoid receptors in anxiety.

Reply. This was already done in response to Comment 7 (see lines 110-123).

Comment No. 16. Clarify whether the 65% increase in anxiety observed in CB1-deficient mice is a statistically significant trend or if individual studies varied widely in their results.

Reply. Sorry for this misunderstanding. I was not sufficiently clear. 65% of all studies (17 out of 26 studies) found that CB1 gene disruption increased anxiety. 31% (8 studies out of 26) reported no effects. I clarified the issue in the new version (line 233).

Comment No. 17. Table 1: Include a brief explanation of the specific anxiety tests used (e.g., CF, EPM, EZM, LD, OF, SI, SPBT) to help readers understand the methods used in the studies.

Reply. Abbreviations were explained in the legend to Table 1 already in the first version (lines 240-242 in the new version). This section was not highlighted because it was present in the original submission.

Comment No. 18. When discussing the variation in findings for CB1-deficient mice, provide possible explanations for the contradictory results (e.g., differences in experimental conditions, testing protocols, genetic background) to shed light on the reasons for the discrepancies.

Reply. Please find the explanations below Table 1 (lines 246-278). I did not change font color, because these explanations were present already in the first version of the manuscript.

Comment No. 19. Provide a clear definition of "aversiveness" in the context of anxiety tests to ensure readers understand its significance in the interpretation of results.

Reply. An overall definition of “aversive conditions” was given in section “3.1. Transgenic animals” where the term was used for the first time (lines 259-267; outlined). Further explanations and examples are given in section “3.3. Enzyme inhibitors and transporters” (lines 430-452) where the “aversiveness hypothesis” is most robustly supported. The second series of explanations (section 3.3.) was not outlined as it was part of the original submission.

Comment No. 20. Expand on the studies that investigated the role of aversiveness in anxiety regulation, highlighting their methodologies and findings. Discuss the implications of these findings for interpreting the anxiolytic effects of CB1 gene disruption under different test conditions.

Reply. As shown in our replies to the previous comment, this information was provided in section 3.3., lines 430-452.

Comment No. 21. What do the subsequent chapters mean?

Reply. The term was discarded. Instead, we indicated the specific section where various issues were presented in detail (see lines 265-267 and 269-270).

Comment No. 22. Summarize the main findings and implications of the discrepancies observed in transgenic animal studies regarding the effects of CB1 and CB2 receptor disruptions on anxiety. Offer insights into potential future research directions that could help resolve conflicting results.

Reply. I believe that the sentence underlined by the reviewer was confusing, therefore I deleted it. This sentence was replaced with several sentences addressing the issues raised by the reviewer (lines 309-315).

Comment No. 23. When discussing the possible existence of a "CB3 receptor" based on early studies, clarify that subsequent research found multiple non-CB1/non-CB2 action sites for cannabinoids, and there is no known CB3 receptor. Emphasize the importance of understanding these non-CB1/non-CB2 targets to explain the divergent pharmacological profile of CB1 ligands.

Reply. I referred to the CB3 receptor in the context of a historical retrospective. The sentences that followed this retrospective clarified the issue, but maybe not sufficiently clearly. I reformulated them to avoid confusions (lines 332-337).

Comment No. 24. Provide additional studies or evidence supporting the concept of a "brain fingerprint" for cannabinoid distribution and its impact on behavior. Discuss how this non-uniform brain distribution can influence anxiety regulation.

Reply. Unfortunately, our study on the “brain fingerprint” of distribution was not followed by others. In addition, the interaction between brain distribution and anxiety-related effects was not studied so far. I had to rely on this single study.

Comment No. 25. Elaborate on the potential challenges and limitations associated with developing drugs targeting specific brain regions and cannabinoid mechanisms for anxiolytic effects. Discuss the complexities of such drug development and the feasibility of achieving desired outcomes.

Reply. In the new version, I addressed the issues raised by the reviewer (lines 386-391).

Comment No. 26. When discussing the initial positive findings with URB-597 and FAAH-KO mice, clarify the role of FAAH in the metabolism of anandamide, leading to increased anandamide levels and potential anxiolytic effects.

Reply. These issues were addressed in response to Comment 10 (lines 179-188).

Comment No. 27. Elaborate on the mechanism by which FAAH inhibition acts as a buffer against environmental aversiveness, leading to anxiolytic effects. Discuss the role of stress and aversive testing conditions in modulating the effects of FAAH inhibition on anxiety.

Reply. The mechanism by which FAAH inhibition can decrease stress-induced anxiety was briefly presented (lines 421-425).

Comment No. 28. When comparing the effects of FAAH and MAGL inhibitors, provide more context regarding the desensitization of central CB1 receptors with chronic MAGL inhibition and the absence of such effects with FAAH inhibitors. This will help readers understand the potential advantages and disadvantages of each approach for anxiolytic drug development.

Reply. The issue was presented in more detail (lines 458-470).

Comment No. 29. Elaborate on the mechanism by which AM-404 affects anxiety, specifically focusing on its role as a transporter blocker and its impact on the availability of endocannabinoids in the brain.

Reply. The issue was already discussed in response to Comment 13 (lines 217-222).

Comment No. 30. Discuss the potential therapeutic applications of FAAH and AM-404 inhibitors in the treatment of psychopathological states induced by traumatic experiences, such as post-traumatic stress disorder (PTSD). Elaborate on the evidence supporting their effectiveness in promoting fear extinction and ameliorating stress-induced anxiety.

Reply. I could expand the review to cover clinical aspects in detail if the reviewer insisted. However, this study is on interventions, mechanisms, and behaviors. Clinical implications were mentioned, but their detailed discussion is outside the scope of the study.

Comment No. 31. Discuss how these coping styles may influence responses to stress and aversive situations.

Reply. We clarified that coping styles revolve around challenge responding, whereas challenges are in fact stress factors and aversive stimuli that need to be handled (lines 499-500).

Comment No. 32. Explain the rationale behind investigating coping styles as a common denominator of anxiety-like behavior and anxiolysis. Discuss how traditional anxiety tests may primarily assess passive coping behaviors, while anxiolytic effects may involve active coping responses.

Reply. It occurs that the text was not sufficiently clear. We believe that coping styles are common denominators of anxiety-like and depression-like behaviors. Passive responses are indicators of both behaviors, whereas active coping is homologous with decreased anxiety and depression. The revised version explains these issues in detail (lines 506-513).

Comment No. 33. Include a brief section discussing human studies that support the link between FAAH gene variants, coping traits, and threat processing in the amygdala. Emphasize the potential relevance of these findings for understanding individual differences in stress response and coping strategies.

Reply. The new version provides more details for the human study that was referred to (lines 532-535). I am not aware other studies of this kind.

Comment No. 34. The statement that "cannabis consumption increases anxiety" needs to be supported by specific references to the studies mentioned in the review (e.g., studies performed over the last 40 years [134-137]). Providing more context and evidence for this claim would enhance the credibility of the interpretation.

Reply. We presented a few studies in more detail to increase the credibility of the statement (lines 543-550).

Minor comments referred to language errors. Reviewer 1 kindly made language corrections, which were observe. In addition, I made a series of language corrections myself. I believe that the language of the article improved considerably.

Reviewer 3 Report

Comments and Suggestions for Authors

Hello:

- This paper is an excellent review of physiological effects of endocannabinoids.

- Thought this paper is meant for a non-clinical audience, looking through a clinical lens, I wish to point out the following:

- What is known about the role of Cannabinoids in treating anxiety in humans? Specifically, THC or CBD? What % do you see the difference at?

- What is known about the role of Cannabinoids in young people, given the explosive numbers of anxiety cases in young people (<25 years).

- Is there a difference in the response to anxiety given synthetic (Sativex) or natural Cannabis.

Author Response

I thank both reviewers for their expert and helpful suggestions.

This reviewer requested the inclusion of more information on the role of natural and synthetic cannabinoid receptor ligands in the clinical treatment of anxiety. I acknowledge that this part of the manuscript was rather superficial. The suggestions of the reviewer were incorporated in the before-the-last paragraph of section “3.2. CB1 receptor pharmacology” (lines 420-452) We added 14 new references to substantiate the content of these paragraphs. Changed text was printed in blue.

My specific replies to the comments are shown below.

Comment No. 1. What is known about the role of Cannabinoids in treating anxiety in humans? Specifically, THC or CBD? What % do you see the difference at?

Reply. We show in the new version that no study showed so far anxiolytic effects for purified THC, whereas the anxiolytic effects of purified CBD were amply confirmed. We also show that cannabis may either increase or decrease anxiety depending on the study. Discrepant findings may be attributed at least partly to the THC:CBD ratio of cannabis preparations. If this ratio was high, the preparation was likely anxiogenic, whereas high CBD content made cannabis anxiolytic. Taken together, this shows that the anxiolytic principle in cannabis is CBD. This, however, is not a primary ligand of CB1 and CB2 receptors; therefore, its role in anxiety control was not discussed in detail. We also show that CBD ameliorates the anxiogenic effects of THC when the two compounds are administered together. We suggest that in addition to the favorable THC:CBD ratio, the absolute dose of THC and the baseline anxiety levels of participants is also important. For instance, CBD ameliorated the effects of THC when the dose of THC was low, but not when the dose was high. The THC:CBD ratio was ~1:1 in both cases. Also, CBD ameliorated the anxiogenic effects of THC when baseline anxiety of participants was low but not when it was high. The THC:CBD ratio was also ~1:1 in this case.

Comment No. 2. What is known about the role of Cannabinoids in young people, given the explosive numbers of anxiety cases in young people (<25 years).

Reply. We showed that cannabis consumption by young people often increases anxiety on the long term; moreover cannabis may contribute to the development of anxiety disorders. This outlines the importance of CBD content, as increasing its concentration in preparations makes cannabis safer.

Comment No. 3. Is there a difference in the response to anxiety given synthetic (Sativex) or natural Cannabis.

Reply. The difference is that the composition of cannabis is highly variable, whereas the stable THC:CBD ratio of Sativex makes it applicable for therapeutic purposes. By ameliorating the anxiogenic effects of THC, CBD may free the way for other therapeutic applications of THC (e.g. glaucoma, cancer); in addition, THC-CBD mixtures proved efficient in reducing anxiety during cannabis withdrawal. These issue were discussed in the new version.

Reviewer 4 Report

Comments and Suggestions for Authors

The review paper is summarizing findings between anxiety and cannnabinoids with relation to stress-related events. The manuscript is based on a lot of studies, which have been done with mouse experiments. The following suggestions will further strengthen the manuscript.

1) Human findings, as well as those with mice, are necessary to describe to put clinical meanings of the manuscript. 

2) Relationships among Stress-Anxiety-Pain-Cannabinoid system with special emphasis on pain should be discussed more on the manuscript for better

 comprehensive undestandings.

3) Peripheral mechanisms of anxiolytic actions through nociceptive afferents should be discussed more.

4) Not only summarized tables but also comprehensive figures are needed for better understandings of the anxiolytic mechanisms.

Comments on the Quality of English Language

None

Author Response

I thank both reviewers for their expert and helpful suggestions. All suggestions were observed. Changed text was printed in blue.

Comment No. 1. Human findings, as well as those with mice, are necessary to describe to put clinical meanings of the manuscript.

Reply. I acknowledge that I neglected human studies, mostly because laboratory findings can shed more light on molecular aspects of the cannabinoid-anxiety relationship than human ones. Yet the latter are also important. To address this issue, I added a paragraph to each of those sections of the manuscript that addressed anxiety. The last paragraph of the section “3.1. Transgenic animals” shows that Complete CB1 disruption was not reported in humans so far, yet polymorphisms in the gene that encodes the CB1 receptor underlie individual differences in anxiety humans, which indirectly supports laboratory findings with knockout animals (lines 318-326). In the before-the-last paragraph of the section “3.2. CB1 receptor pharmacology” I discuss the effects of Δ9-THC, cannabidiol and their combination on anxiety in humans. I chose these compounds because they are widely consumed, for which data are abundant (lines 420-452). In the last paragraph of section “3.3. Enzyme inhibitors and transporters” I present human findings on the anxiety related effects of FAAH polymorphisms and FAAH antagonists. These entirely support laboratory findings (lines 543-560). Finally, I highlighted the last paragraph of section “3.4. FAAH inhibitors and coping styles” where I presented data on the impact of cannabinoid signaling on coping styles in humans (lines 602-608).

Comment No. 2. Relationships among Stress-Anxiety-Pain-Cannabinoid system with special emphasis on pain should be discussed more on the manuscript for better comprehensive undestandings.

Reply. This relationship was addressed in section “3.2. CB1 receptor pharmacology” (paragraph lines 404-407 and 445-451) and in section “3.3. Enzyme inhibitors and transporters” (lines 556-560). Please note however, that this review focuses on molecular aspects of the anxiety-cannabinoid relationship. As such, interactions between various diseases cannot be discussed in detail here.

Comment No. 3. Peripheral mechanisms of anxiolytic actions through nociceptive afferents should be discussed more.

Reply. We addressed this issue in the section “3.2. CB1 receptor pharmacology” (lines 408-418). We showed that nociceptive afferents influence the function of brain regions involved in affective and cognitive processes, and that chronic pain often elicits the development of anxiety. In fact, pain-induced anxiety is a specific case of anxiety associated with stress. Not surprisingly, cannabinoid receptor ligands decrease pain-induced anxiety more robustly as compared to other types of anxiety. In subsequent parts of the manuscript, we developed the idea further as shown in our replies to Comment No. 2.

Comment No. 4. Not only summarized tables but also comprehensive figures are needed for better understandings of the anxiolytic mechanisms.

Reply. Indeed, visually transmitted information may help the reader. To address this comment, we compiled Fig. 1 to illustrate the diversity of mechanisms activated by cannabinoid receptor ligands, and Fig. 2 to compare the three main research approaches regarding the mechanisms activated.